# A Sea Anemone *Lebrunia neglecta* Venom Fraction Decreases Boar Sperm Cells Capacitation: Possible Involvement of HVA Calcium Channels

**DOI:** 10.3390/toxins14040261

**Published:** 2022-04-07

**Authors:** Fernando Lazcano-Pérez, Karina Bermeo, Héctor Castro, Zayil Salazar Campos, Isabel Arenas, Ariana Zavala-Moreno, Sheila Narayán Chávez-Villela, Irma Jiménez, Roberto Arreguín-Espinosa, Reyna Fierro, Humberto González-Márquez, David E. Garcia, Judith Sánchez-Rodríguez

**Affiliations:** 1Instituto de Ciencias del Mar y Limnología, Universidad Nacional Autónoma de México, Puerto Morelos, Quintana Roo 77580, Mexico; ferlaz@hotmail.com (F.L.-P.); sheila_chavezuaslp@outlook.es (S.N.C.-V.); 2Departamento de Fisiología, Facultad de Medicina, Universidad Nacional Autónoma de México, Ciudad de Mexico 04510, Mexico; karbemocspsi@gmail.com (K.B.); hecas006@comunidad.unam.mx (H.C.); isabel.arenas@unam.mx (I.A.); erasmo@unam.mx (D.E.G.); 3Facultad de Ingeniería, Universidad Nacional Autónoma de México, Ciudad de Mexico 04510, Mexico; zayil.salazar@gmail.com; 4Departamento de Ciencias de la Salud, Div. C.B.S., Universidad Autónoma Metropolitana-Iztapalapa, Ciudad de Mexico 09310, Mexico; jimi@xanum.uam.mx (I.J.); reyna@xanum.uam.mx (R.F.); hgm@xanum.uam.mx (H.G.-M.); 5Facultad de Ciencias, Universidad Nacional Autónoma de México, Ciudad de Mexico 04510, Mexico; anairazm@ciencias.unam.mx; 6Departamento de Química de Biomacromoléculas, Instituto de Química, Universidad Nacional Autónoma de México, Ciudad de Mexico 04510, Mexico; arrespin@unam.mx

**Keywords:** *Lebrunia neglecta*, sea anemone, boar sperm, capacitation, HVA calcium channels, chromaffin cells

## Abstract

Sea anemones produce venoms characterized by a complex mixture of low molecular weight compounds, proteins and peptides acting on voltage-gated ion channels. Mammal sperm cells, like neurons, are characterized by their ion channels. Calcium channels seem to be implicated in pivotal roles such as motility and capacitation. In this study, we evaluated the effect of a low molecular weight fraction from the venom of the sea anemone *Lebrunia neglecta* on boar sperm cells and in HVA calcium channels from rat chromaffin cells. Spermatozoa viability seemed unaffected by the fraction whereas motility and sperm capacitation were notoriously impaired. The sea anemone fraction inhibited the HVA calcium current with partial recovery and no changes in chromaffin cells’ current kinetics and current–voltage relationship. These findings might be relevant to the pharmacological characterization of cnidarian venoms and toxins on voltage-gated calcium channels.

## 1. Introduction

Sea anemones produce venoms characterized by a complex mixture of low molecular weight compounds, peptides, and proteins [1,2]. Some of their neurotoxic peptides that act on voltage-gated sodium and potassium channels have been extensively studied as molecular tools and potential therapeutic agents [3]. For instance, ShK, a potent K_V_ channel toxin isolated from *Stychodactyla helianthus*, has been proposed as a potential therapeutic for multiple sclerosis treatment since it potently blocks lymphocytes’ K_V_1.3 channel [4]. In an effort to find new potential drugs from marine sources, some sea anemone venoms have also been screened for anticancer and antimicrobial activity, showing significant results [5,6]. Hence, considering the variety of bioactive compounds that sea anemones produce, we might expect that research on their venoms and toxins could be useful in novel pharmacological and therapeutic targets, particularly on ion channels.

Sea anemone toxins that affect voltage-gated ion channels have been studied in several types of excitable cells such as neurons, HEK cells and cardiomyocytes. Mammal sperm cells are characterized by their ion channels and membrane receptors, implicated in vital processes such as capacitation and acrosomal reactions [7]. Capacitation is the process that allows sperm cells to acquire fertilizing capacity and happens through a series of molecular events that are reflected in the sperm cells as physiological changes. For example, the membrane fluidity increases as a result of cholesterol efflux, and there is an increase of calcium permeability and hyperpolarization, with a consequent increase of pH and protein tyrosine phosphorylation; all these changes result in the modification of the sperm motility pattern [8,9]. Ion channels seem to be at the center of all these molecular mechanisms, and their role in hyperactivation, capacitation and chemotaxis has been demonstrated by pharmacological, molecular and electrophysiological studies; among these channels, we can find Na_V_, K_V_, Ca_V_ channels and TRP family receptors [10]. Calcium channels are particularly essential for sperm cell function [11].

Calcium channels are classified into two main classes: high voltage activated (HVA) and low voltage activated (LVA). HVA channels are subdivided according to the behavior of their currents into L, N, P/Q and R types, and open after strong depolarizations. In contrast, LVA channels open after weak depolarizations and have been termed as T-type [12]. L, T and P/Q type calcium channels have been identified in the sperm cells of many mammal species [13]. Sperm cells also contain a unique ion channel known as CatSper (cation channel of sperm) which plays crucial role in fertility and motility [11].

The research one the effect elicited by toxins from animal venoms on sperm cells has scarcely been addressed. For instance, two toxins from the scorpion *Parabuthus granulatus* decrease T-type Ca^2+^ channel activity in mouse spermatogenic cells and also inhibit the acrosome reaction in mature sperm [14]. Romero and collaborators [15] described an increase in the concentration of intracellular Ca^2+^ in spermatozoa upon black widow spider *Latrodectus mactans* venom application. Later, the same group found that this effect is due to the negative impact of TEA-sensitive K^+^ current in sperm cells [16]. Escoffier and collaborators screened the venom of several snakes and found that a low molecular weight fraction of the venom from the elapid *Oxyuranus scutellatus scutellatus* contains two phospholipases A_2_ that trigger sperm acrosome reaction by inducing lipid rearrangements of the sperm cell plasma membrane [17]. To our knowledge, cnidarian toxins have not been explored in this regard.

*Lebrunia neglecta,* (formerly *L. danae*) is a sea anemone of the family Aliciidae that inhabits shallow waters around coral reefs along the entire Caribbean Sea [18]. Its venom is very irritant and it has been reported as one of the most venomous sea anemones in the Caribbean region [19]. Some experiments on different models have shown its cyto- and neuro-toxicity [20]. Previous studies have shown the existence of Ca_V_ channel toxins in other anthozoan venoms [21]. In the present study, we assess the effect of a low molecular weight fraction of *L. neglecta* venom in boar sperm cells and L-type Ca_V_ channels of chromaffin cells to explore sea anemones as a new source of potential pharmacological agents targeting calcium channels.

## 2. Results and Discussion

### 2.1. Venom Extraction and Fractionation

Sea anemone toxins acting on ion channels are low molecular weight peptides; hence, size exclusion chromatography was performed to eliminate high molecular weight proteins from the venom. This procedure resulted in three fractions (Figure 1A). Fraction 3 (F3) showed marked activity on sperm cells. The MALDI-TOF mass spectrometry analysis of this fraction confirmed the existence of peptides between 1–7 kDa (Figure 1B).

### 2.2. Sperm Viability

Sperm cells are highly specialized eukaryotic cells that react to any changes in their physiology in short time lapses, so they are of particular interest when studying disruption by toxins [22]. Boar sperm cells have been extensively used in biomedical research because of their physiological similarity to human sperm cells, and have been used in many studies that include renal hypertension, atherosclerosis and reproductive pathologies [23]. These cells are also a very accessible model, since their motile nature and physiological processes are tightly linked to ion channel functioning.

According to our results, F3 decreases sperm viability at 13.8% and 18.4% in the presence of 20 µg protein/mL and 50 µg protein/mL of F3, respectively (Figure 2).

Since the viability of the spermatozoa showed a minor effect after F3 exposure, the next aim was to evaluate motility.

### 2.3. Sperm Motility

After 4 h of incubation with 10 μg protein/mL of F3, sperm motility was affected. A few clumps of agglutinated sperm cells that still showed flagellar movement appeared. As the concentration of F3 increased (20 μg protein/mL and 50 μg protein/mL), the number of free sperm cells with progressive motility decreased, the clumps became denser and the flagellar movement inside the clumps diminished (Table 1). Motility is directly linked to the ability of sperm cells to fertilize and it can be impaired by several exogenous substances such as lipopolysaccharides, hemolysins and lectins [24]. Toxins from starfish have also been demonstrated to cause immobilization and adhesion in sea urchin sperm cells [25].

Our results show that boar sperm agglutination increases whereas sperm motility decreases in a dose-dependent way (Table 1, Figure 3). Few studies describe the agglutination effect of exogenous substances on sperm cells. The origin of this agglutination could be explained by the presence of antigens in the sperm cell membrane produced by contact with the peptides contained in the venom fraction, since the surface of spermatozoa is rich in glycoproteins that function as receptors for agglutination factors (e.g. the sperm agglutination factor from *E. coli*) [26]. Lectins have also been reported in sea anemone venoms [27]; however, lectins possess higher molecular weights than those detected in our MALDI-TOF analysis. The nature of the molecule and process causing the sperm agglutination remains to be elucidated until a complete chemical characterization of the fraction has been conducted.

### 2.4. Capacitation

Sperm capacitation was evaluated by the CTC staining method. As shown in Table 2, capacitation decreased in a dose-dependent manner in the presence of F3. This result could be due to a decrease in external Ca^2+^ influx [8] or internal Ca^2+^ movement [28], since it is known that Ca^2+^ ions are essential in sperm hyperactivation, and also in sperm detachment from agglutination.

Several ion channels are essential components of mammalian sperm cells capacitation, such as K_V_ and Ca_V_ channels and TRPC receptors. In particular, calcium is crucial for processes related to fertilization, such as motility and capacitation [29,30]. Sperm cells express the CatSper channel, one of the main channels required for motility, along with HVA calcium channels. The latter has been related to the modulation of CatSper activity, acrosome reaction and motility [13]. Since one of the purposes of this paper is to know whether F3 produces an effect on motility and capacitation of sperm cells by affecting HVA calcium channels, we tested the fraction on a model with a high expression of this channel population, such as the chromaffin cells of rats [31].

### 2.5. Effect of F3 on Calcium Current Amplitude

Calcium currents were elicited with a square pulse from −80 to 0 mV every 5 s. Figure 4A shows representative current traces under Control, F3 and Cd^2+^ conditions. F3 significantly inhibited calcium current density, measured isochronally at 50–60 ms, starting test pulse to 0 mV (Figure 4D) (z = 28, *n* = 7, *p* ≤ 0.05). This effect was partially reversed after washout as it was observed in the time course of current amplitude (Figure 4B). Traces in the control and F3 condition were normalized and superimposed, showing no changes in current kinetics (Figure 4C), suggesting a voltage-insensitive mechanism involved in the inhibition of the current amplitude. Moreover, an effect of F3 in a concentration-dependent manner was observed (Figure 4E). Data points were fitted with Equation (1) to obtain an IC_50_ = 18.08 ± 6.2 µg/mL of protein and a Hill coefficient = 0.56 ± 0.1. In summary, F3 produces an inhibition of the calcium current with partial recovery and no changes in the current kinetics. The effect of anemone venoms on ion channel activity has been documented in sodium [32,33] and potassium channels [34]. However, little is known about the effect of anemone venoms on calcium channels. A previous work reported a similar effect of another anemone venom on HVA calcium channels [21].

### 2.6. F3 Inhibits Calcium Currents without Changes in Voltage Dependence

Another aim of this work was to explore whether or not F3 produces changes in the voltage dependence of calcium channels. To evaluate the effect of F3 on voltage-dependent behavior, an I–V curve was obtained with voltage steps from −80 to +60 mV. No modifications in voltage dependence were found (V_½_ = −21.08 mV and −23.3 mV for Control and F3, respectively; V_rev_ = 49 mV and 47 mV for Control and F3, respectively). On the other hand, a statistical inhibition at −40 and 0 mV was found (F_(14,126)_ = 3.41, *n* = 9, *p* ≤ 0.05). These data were confirmed with a ramp protocol, as it was evoked from −80 to +60 mV (0.7 mV/ms), as shown in Figure 5B. The maximal peak current was reduced in the presence of F3. In addition, only one component was observed in the ramp trace, which agrees with previous reports showing that HVA is the major component in chromaffin cells [31]. Following the I–V relationship, the tail current amplitude was reduced in presence of the fraction. Additionally, a conductance curve was obtained from the tail currents and the data points were fitted to Equation (2). No changes were observed in this property (Figure 5C). F3 did not affect the voltage dependence; therefore, it inhibits calcium current influx by reducing the number of ion channels without kinetic changes. These results support the fact that F3 may produce changes on the surface charge of the cell membrane or on the voltage sensor. Recently, it has been suggested that the mechanism on calcium channels from *Palythoa caribaeorum* venom, another cnidarian, produces a similar effect on calcium channel current without modifications in the voltage dependence [21].

L-type calcium channels are highly expressed in sperm, particularly Ca_V_1.2 and Ca_V_1.3 [11,35]. Consistently, their inhibition by dihydropyridines interrupts acrosome reaction and other sperm functions [36]. These channels are the same type expressed in chromaffin cells [37]; therefore, the latter model is appropriate to study the effect of F3 on L-type calcium channels.

### 2.7. F3 Does Not Inhibit L-Type Calcium Channels Only

To find out if F3 specifically inhibits L-type calcium channels, we employed the maximum nifedipine concentration (Nif), a selective L-type calcium channel blocker [38,39]. As shown in Figure 6A, 5 µM Nif inhibits the calcium current amplitude by 40.5 ± 4.7% (*n* = 11) with a partial recovery (Figure 6B). This result suggests that almost 50% of the population of calcium channels are L-type, coinciding with previous works [31]. Moreover, the inhibition by Nif was greater than the inhibition by F3 because of the concentration used (10 µg/mL), which does not produce the maximal effect (Figure 4E). However, when Nif and F3 were applied together, the amplitude of the current was reduced further compared to Nif only (54.7 ± 2.2%, *n* = 3) (Figure 6C,D), suggesting that the L-type is not the unique target. These results are statistically significant (X^2^ = 10.8; *p* ≤ 0.05), with differences between Nif, Nif + F3 and F3 (Figure 6E). The results suggest that F3 also inhibits another type of HVA calcium channel.

## 3. Materials and Methods

### 3.1. Venom Extraction and Fractionation

*Lebrunia neglecta* organisms were collected at Puerto Morelos Reef Lagoon, by SCUBA diving at approximately 4 m depth. The sea anemones were kept in plastic bags with sea water, and carried to the laboratory. The extraction was conducted in a Ten Broeck homogenizer (Pyrex^®^) using distilled water. Nematocyst discharge was monitored with a microscope in order to ensure the obtention of their active compounds. The crude extract was fractionated by size exclusion chromatography in a 118 × 1.5 cm Sephadex^®^ G-50 (Pharmacia Biotech) column using 0.3 M acetic acid as eluant at a 3 mL/min flux. Fractions of 15 mL were collected in conical tubes (Corning^®^) and the absorbance was monitored at 280 nm using an Äkta prime plus chromatographer. The tubes corresponding to the same fraction were pooled and concentrated by reduced pressure in a Büchi R-215 rotary evaporator. Finally, the three fractions F1, F2 and F3 were lyophilized in a Labconco™ freeze dry system and stored at −70 °C until use.

### 3.2. Protein Quantification

Protein concentration was measured by the Bradford method with Quick Start™ (BIORAD) kit using bovine serum albumin (BSA) in a 96-well plate. Absorbance was monitored at 595 nm with a Stat Fax 4200^®^ plate reader.

### 3.3. Mass Spectrometry Analysis

The molecular mass range of the peptides was determined by MALDI-TOF mass spectrometry. 5 μL of a saturated solution of 4-hydroxy-α-cyanocinnamic acid was added to 5 μg of sample. 1 μL of this solution was deposited onto the MALDI plate and allowed to dry at room temperature. The spectrum was recorded in the positive linear mode on a Bruker Microflex mass spectrometer equipped with a nitrogen laser at 337 nm.

### 3.4. Collection and Semen Sample Preparation

Semen samples were obtained from a sperm-rich ejaculated fraction of healthy, fertile boars. The samples were examined for their motility and viability and classified as normozoospermic, according to criteria described by Garner (2000) [40] and Jimenez (2003) [7]. Sperm viability was evaluated by eosin-nigrosin staining: a drop of 5 µL of sperm sample, previously washed, was added in a glass slide, mixed with 5 µL of eosin-nigrosin, smeared, and analyzed under phase-contrast microscopy. Two hundred cells were observed to evaluate the semen samples. Samples with viability and motility >80%, abnormalities <15% and a concentration >80 × 10^6^ spermatozoa/mL were used.

Agglutination was evaluated by phase-contrast microscopy. 10 μL of the sample was placed on a slide and observed at 100× magnification, counting 10 different fields. A semiquantitative classification of agglutinated sperm was performed, where + corresponds to 10% sperm forming clumps, ++ corresponds to around 30% of clumps, +++ corresponds to 60% of the cell population is agglutinated, ++++ corresponds to 80% of the cells in clumps of different size, and +++++ corresponds to ≈ 95% of the cells in clumps.

### 3.5. Capacitation

Semen samples were washed twice to remove the seminal plasma. Then, 1 mL of phosphate-buffered saline (PBS) was added to an equal volume of semen, followed by centrifugation at 600× *g* for 5 min. Aliquots with 5 × 10^6^ cells were placed on a Nunc 4-Well Dishes (Nunc, Roskilde, Denmark), with HEPES-TALP medium, supplemented with 6 mg/mL BSA fraction V and 1 mM sodium pyruvate, and incubated for 4 h in a humidified atmosphere of 95% air/5% CO_2_ at 38 °C. Capacitation was assessed by the chlortetracycline (CTC) staining method. Briefly, 5 µL of spermatozoa was mixed with 5 µL of CTC solution, containing 750 µM CTC, 130 mM NaCl, 5 mM L-cysteine and 20 mM Tris HCl, at pH 7.8. Spermatozoa were fixed with 5 µL of 0.2% glutaraldehyde in 0.5 M Tris buffer, at pH 7.4. Samples were mounted on microscope slides with glass coverslips. Slides were observed under a fluorescence microscope (Olympus BX51, Japan) at 495 nm UV epifluorescence and 400× magnification. Capacitation was determined as follows: non-capacitated spermatozoa presented fluorescence throughout the head and capacitated spermatozoa showed intense fluorescence in the equatorial and acrosomal zone [41].

### 3.6. Toxin Exposure

The lyophilized F3 was resuspended in the capacitation medium to obtain a stock solution of 1 mg/mL. Semen samples were incubated in a Nunc 4-Well Dishes (InVitro^®^) for 4 h in the presence of the F3 at concentrations of 10, 20 and 50 μg protein/mL. Viability and motility were assessed under a stereomicroscope (NIKON) when the incubation period ended.

### 3.7. Chromaffin Cells Culture

Chromaffin cells were obtained from 10-day-old male Wistar rats as previously described (Morgado-Valle et al. 1998). Rats were anesthetized by intraperitoneal injection of pentobarbital sodium, and adrenal glands were removed and transferred to a Hank´s Balanced Salt Solution (HBS). The adrenal medulla was isolated and incubated at 37 °C for 30 min in HBS plus 2 mg/mL collagenase type I (Sigma, St. Louis, MO, USA) and 15 mg/mL deoxyribonuclease I (Sigma, St. Louis, MO, USA). After enzymatic digestion, the tissue was mechanically dispersed and the cell suspension was centrifuged at 180× *g* for 7 min and washed twice in Dulbecco´s modified Eagle´s medium (Life Technologies, Carlsbad, CA, USA), supplemented with 10% fetal bovine serum (Life Technologies, Carlsbad, CA, USA), 4.5 µg/mL insulin (Sigma; St. Louis, MO, USA), 1% penicillin-streptomycin (Life Technologies, Carlsbad, CA, USA), and 2.5 µg/mL fungizone (Life Technologies, Carlsbad, CA, USA). Cells were plated on polystyrene culture dishes coated with poly-L-lysine and stored in a humidified atmosphere of 95% air/5% CO_2_ at 37 °C. Cells were studied 72–96 h after plating.

### 3.8. Electrophysiological Recording

Calcium currents were recorded, employing the patch-clamp technique in whole-cell configuration with an EPC-9 amplifier (HEKA Elektronik GmbH; Lambrecht/Pfalz, Germany) at room temperature. Currents were sampled at 10 kHz and filtered at 2.9 kHz. Electrodes were made of borosilicate glass capillary tubes (Kimble Chase, Vineland, NJ, USA) with a resistance of 3–4 MΩ. The series resistance (3–10 MΩ) was compensated >50%. Cells were continuously bathed with control or test solutions at a rate of 2 mL/min. Barium was used as a charge carrier in solutions designed to minimize undesired currents and increase the current through calcium channels. The bath solution contained (in mM): 120 NaCl, 10 BaCl_2_, 2 MgCl_2_, 10 glucose, 10 HEPES and 0.0002 TTX; pH was adjusted at 7.4 with NaOH. The pipette solution contained (in mM): 130 CsCl, 4 MgCl_2_, 10 EGTA, 10 HEPES and 3 Na_2_ATP; pH was adjusted at 7.2 with CsOH. Calcium channel current was defined as the component of the sensitive current to 100 µM CdCl_2_. Pharmacological identification of L-type calcium channels was performed using Nif (Sigma Aldrich) at 5 µM in the bath solution. F3 was stored at −20 °C in a stock solution in distilled water and diluted to 0.1, 1, 10, 100, and 1000 µg/mL in the bath solution before using it. F3 dilutions were applied by a perfusion system (ValveBank8, Automate Scientific, Berkeley, CA, USA), except for the 1000 µg/mL dilution, for which an Eppendorf 5646 transjector (Eppendorf, Madison, WI, USA) was used.

### 3.9. Statistical Analysis

Results are represented as mean ± standard error of the mean (SEM). Statistically significant differences between control and F3-treated cells were analyzed by the Wilcoxon test, due to the non-normal distribution. For three or more comparisons, one-way analysis of variance (ANOVA) by repeated measures was employed to measure differences in control, F3 and Nif application. The Friedman test was used for the data with non-normal distribution, followed by Sidak’s post hoc in repeated measures, or the Kruskal–Wallis test followed by Dunn’s post hoc in not pairing measures. For I–V comparisons, two-way ANOVA was performed followed by Sidak’s post hoc. Statistical significance was taken at *p* ≤ 0.05. Graphics and analysis were constructed using IGOR Pro (Version 6.1.2.1, WaveMetrics, Lake Oswego, OR, USA) and GraphPad Prism 6.0 (GraphPad Software, San Diego, CA, USA). The following equations were fitted:

The concentration-response curve was fitted with the Hill equation (Equation (1)):(1)$$R=base+(\max-base\;)/\left\{1+\left[{\frac{IC_{50}}{C}}^{H}\right]\right\}$$
where *R* is the current inhibition, *C* is the test concentration, *IC_50_* corresponds to the F3 concentration giving half of the maximal inhibitory response, and *H* is the Hill coefficient.

The conductance curve was fitted with a single Boltzmann equation (Equation (2)):
(2)$$I/I_{max}=(1+exp{\left(\left(V_{1/2}-V_{m}\right)/\left(k\right)\right)}^{-1}$$
where *I/I_max_* is the normalized current amplitude at test potential *V_m_*, *V_½_* is the voltage of half-maximal activation and inactivation, and *k* is the slope factor.

The current–voltage (I–V) relationship was fitted with a modified Boltzmann equation (Equation (3)):
(3)$$I=\left(V_{m}-V_{rev}\right)*g_{max}/\left(1+\exp\left(\frac{V_{1/2}-V_{m}}{k}\right)\right)$$
where *I* is the current amplitude measured at the test potential *V_m_*, *V_rev_* is the reversal potential, *g_max_* is the maximum slope conductance, *V_½_* is the half-activation potential, and *k* is the slope factor.

## 4. Ethical Statement and Animal Care

Male Wistar rats were obtained from the animal breeding facility of the School of Medicine at Universidad Nacional Autónoma de México (UNAM). The rats were transported in plexiglass cages and were sacrificed immediately. All animals were handled according to the guidelines and requirements of the National Institutes of Health Guide for the Care and Use of Laboratory Animals (8th edition) and the Mexican Official Norm for Use, Care and Reproduction of Laboratory Animals (NOM-062-ZOO-1999). Experimental protocols were reviewed and approved (identification code: FM/DI/059/2018—approved on 28 June 2018) by the Committee of Research and Ethics of the School of Medicine, UNAM.

## 5. Conclusions

Venoms and extracts from anthozoans such as soft corals, zoanthids and sea anemones have been explored as a source of bioactive molecules with potential pharmacological activity. Most of this research has been focused on finding anticancer, antibiotic compounds and ion channel modulators. Focus on the latter has mainly been on the Na_V_ and K_V_ channels of neuronal and cardiac cells. This study explores the effects of cnidarian venoms and toxins on the sperm cell model and reinforces the evidence of the existence of Ca_V_ channel toxins in cnidarian venoms. These results extend the possible application of sea anemone toxins as pharmacological tools.

## Figures and Tables

**Figure 1 toxins-14-00261-f001:**
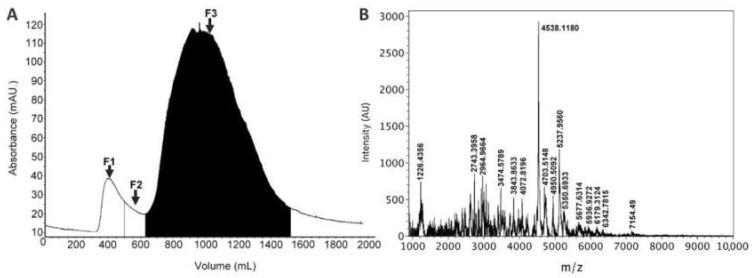
(**A**) Gel filtration chromatogram from the separation of *Lebrunia neglecta* venom on a Sephadex G50 column. (**B**) MALDI-TOF mass spectrum of F3.

**Figure 2 toxins-14-00261-f002:**
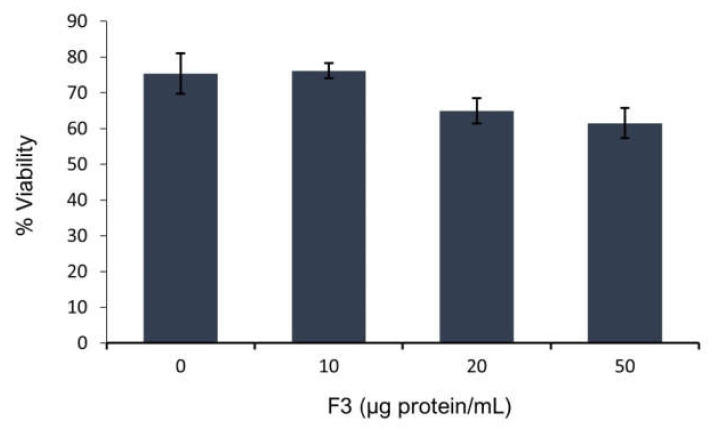
Sperm viability in presence of F3 at different concentrations after 4 h of incubation. Sperm viability was analyzed using the eosin-nigrosin staining. Initial viability at time zero was 82%. Values are expressed as mean ± SEM (*n* = 3).

**Figure 3 toxins-14-00261-f003:**
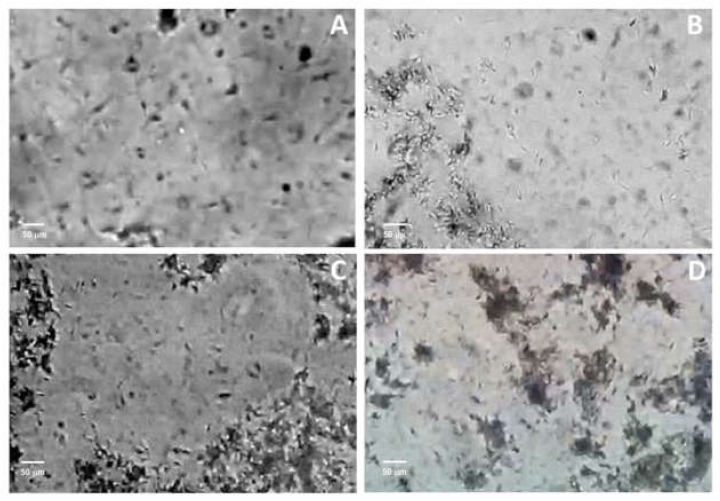
Boar sperm cells agglutination by F3. (**A**) Control. (**B**–**D**) Sperm cells after exposure to F3 at 10, 20 and 50 μg protein/mL concentrations, respectively.

**Figure 4 toxins-14-00261-f004:**
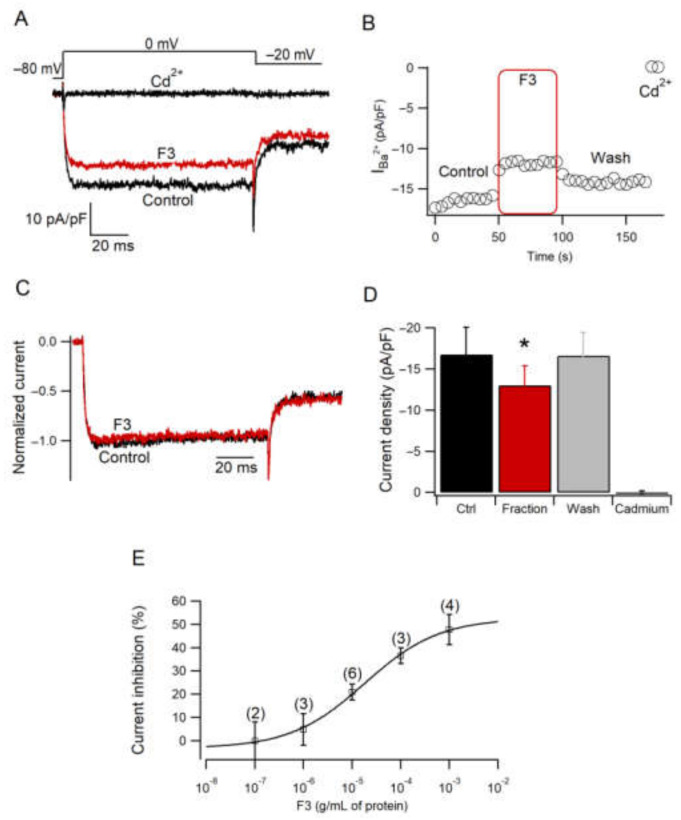
Effect of F3 on calcium current amplitude. (**A**,**B**) Representative traces and time course of calcium current in Control, F3 and Cd^2+^ conditions obtained by the voltage protocol shown at the top of the trace in A. (**C**) Representative normalized calcium current traces in Control and F3 conditions. (**D**) Current density is presented as the mean ± SEM for each condition. Data were analyzed by Wilcoxon test (z = 28; *n* = 7; * *p* ≤ 0.05). (**E**) Dose-response curve for the inhibition of calcium current with different concentrations of F3. Each point represents the mean ± SEM; n is indicated for every point. The line represents the best fitting of the Hill equation to the data (Equation (1)).

**Figure 5 toxins-14-00261-f005:**
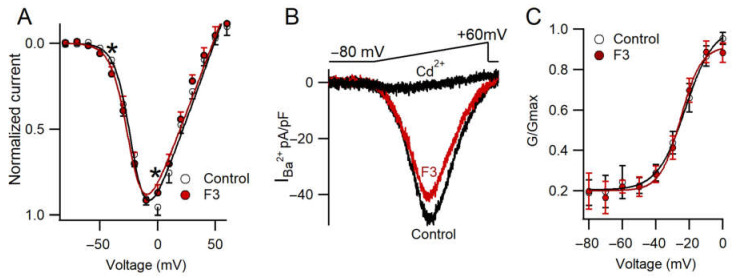
F3 inhibits calcium current without changes in the voltage dependence. (**A**) I–V relationship for Control and F3 conditions. Each point represents the mean ± SEM. The solid line represents the best fit to a single Boltzmann equation (Equation (3)). Data were analyzed by two-way ANOVA (F_(14,126)_ = 3.41, *n* = 9, * *p* ≤ 0.05). (**B**) Representative current elicited by a voltage-ramp protocol under Control, F3 and Cd^2+^ conditions. (**C**) Conductance curve in Control (white circles) and F3 condition (red circles). Each point represents the mean ± SEM. The solid line represents the best fit to a single Boltzmann equation (Equation (2)).

**Figure 6 toxins-14-00261-f006:**
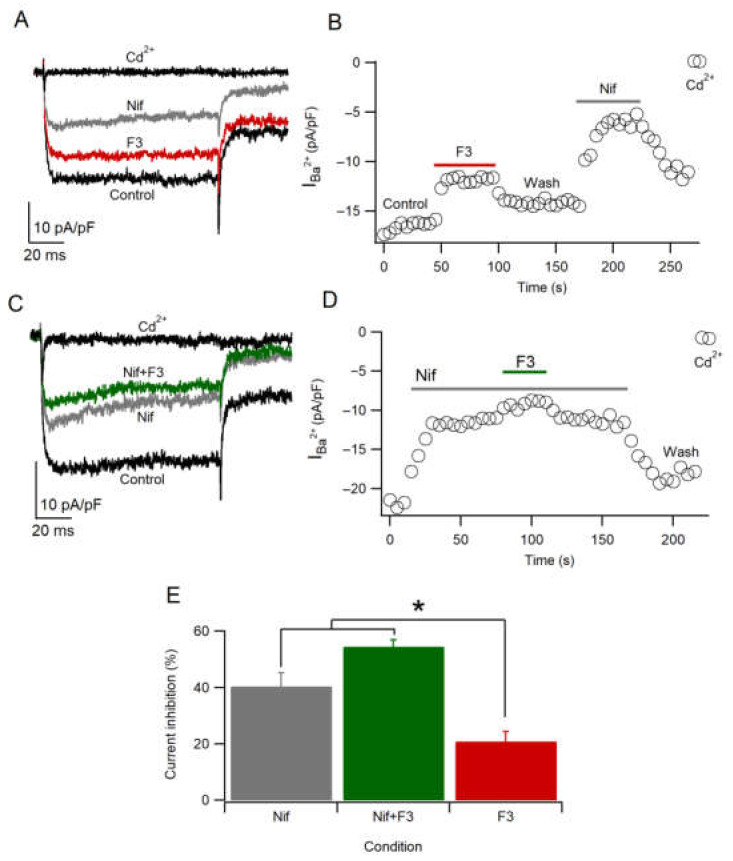
F3 does not inhibit L-type calcium channels only. (**A**,**B**) Representative traces and time course of calcium current in Control, F3, Nif and Cd^2+^ conditions. (**C**,**D**) Representative traces and time course of calcium current in Control, Nif, Nif + F3 and Cd^2+^ conditions. (**E**) Calcium current inhibition represents the mean ± SEM in presence of Nif (*n* = 11), Nif + F3 (*n* = 3) and F3 (*n* = 6) conditions. Data were analyzed by Kruskal–Wallis test (X^2^ = 10.8; * *p* ≤ 0.05).

**Table 1 toxins-14-00261-t001:** Sperm cells’ motility and agglutination after 4 h of incubation with F3.

Parameter	F3 (μg Protein/mL)			
	0.0	10	20	50
Motility	+++++	++++	++	+
Agglutination	++	++	++++	+++++

Motility and agglutination were measured by a semiquantitative characterization. Linear motility: +++++ ≈ 95%, ++++ ≈ 70%, ++ low and vibratory movements, + zero. The size and number of clumps were used to measure agglutination. Cells in clumps: +++++ ≈ 95%, ++++ ≈ 80%, ++ ≈ 30%, + ≈ 10%.

**Table 2 toxins-14-00261-t002:** Sperm capacitation in the presence of F3.

F3 (μg Protein/mL)	Spermatozoa Status (%)	
	Non-Capacitated	Capacitated
0.0	32.92 ± 3.54	63.41 ± 6.36
10	62.5 ± 1.41	35.60 ± 9.19
20	79.11 ± 1.41	18.22 ± 3.54
50	96.47 ± 5.66	2.94 ± 2.12

Non-capacitated sperm cells and capacitated cells after 4 h of incubation and CTC staining. Values are expressed as mean ± SEM (*n* = 3).

## Data Availability

Not applicable.

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
