# Peer review of "A Sea Anemone Lebrunia neglecta Venom Fraction Decreases Boar Sperm Cells Capacitation: Possible Involvement of HVA Calcium Channels"

_toxins, 2022, doi:10.3390/toxins14040261_

Round 1
Reviewer 1 Report
A brief summary:
The present study is the work on the venom of sea anemones. The venom is a low MW compound and acts on voltage-gated ion channels. The researcher used sperm and chromaffin cells.
General concept comment:
The work is very interesting because this was the first venom from Lebrunia neglecta and was characterized for Ca channels.
The method is clear and well described.
The experiments were prepared very well with the rational and easy to follow.
It would be more intriguing if the researchers performed LC-MS/MS for the venom to characterize the venom formula.
To determine the sperm viability, how many sperms (cells) were done in each test? What is the standard error in Table 1?
In the sperm motility test, How many replicates were done?
In the capacitation test, How many replicates were done? Which statistic was used in Table 3.
Because this F3 is the crude extract, Was 15 ml of collection (from method section) used in all experiments?
Author Response
Reviewer #1:
- It would be more intriguing if the researchers performed LC-MS/MS for the venom to characterize the venom formula.
Yes, the purification of the peptides and the elucidation of their amino acid sequence would be optimal. This study is a preliminary on a species never studied before. We are in the process of complete chemical characterization of this fraction and the rest of the venom.
- To determine the sperm viability, how many sperms (cells) were done in each test? What is the standard error in Table 1?
We apologize for the omission of the statistics here. It has been corrected in text and figure caption. The presentation was changed to a graph as suggested by another reviewer.
- In the sperm motility test, How many replicates were done?
The statistics has been added.
- In the capacitation test, How many replicates were done? Which statistic was used in Table 3.
The data have also been added.
- Because this F3 is the crude extract, Was 15 ml of collection (from method section) used in all experiments?
F3 is not the crude extract; it is the third fraction obtained after a size exclusion chromatography of the crude extract. During this chromatographic procedure, the chromatographer was programmed to collect fractions of 15 mL in conical tubes until the extract was separated in the three fractions mentioned. The tubes that corresponded to the same fraction were pooled. We added more information in the methods section to clarify the procedure a little bit more and avoid confusion.

Reviewer 2 Report
This work presents interesting activity of the sea anemone venom extract on sperm cells. However, this is very preliminary. The following experiments are recommended:
1) MALDI mass analysis of F3 indicates the presence of many components thus it is recommended to purify and characterize the component responsible for activity.
2) Is the activity specific to Cav1.2 or Cav1.3. This is very important for calcium channel pharmacology if the component/s responsible for activity is selective of specific to one Cav subtype since very few known compounds exhibit such specificity/selectivity.
Author Response
Reviewer #2:
- MALDI mass analysis of F3 indicates the presence of many components thus it is recommended to purify and characterize the component responsible for activity.
Yes, we agree with the reviewer that the purification and characterization of the compound or compounds responsible for the biological activity will be of great interest and must be done. We are in the process of obtaining data about the chemical structure of the components in the fractions. This information will be published in another paper.
- Is the activity specific to Cav1.2 or Cav1.3. This is very important for calcium channel pharmacology if the component/s responsible for activity is selective of specific to one Cav subtype since very few known compounds exhibit such specificity/selectivity.
As shown in Fig. 5, F3 affects a fraction of the nifedipine-sensitive CaV and an additional fraction of the total Cd2+ sensitive voltage-gated CaV current; therefore, only a non-specific action of F3 can be supported until this stage of the experiments. Future experiments should be performed using purified peptides from F3 to support a specific CaV target.

Reviewer 3 Report
The authors herein reported the effect of bio-guided fractionations of l.negletta venom on dorm cells. The experimental detail are finely described. Results are well showed and data support the conclusion. Only a point should be better described. Why do not continue to affectionate f3 using other strategies?
Author Response
Reviewer #3
- The authors herein reported the effect of bio-guided fractionations of L. neglecta venom on dorm cells. The experimental detail are finely described. Results are well showed and data support the conclusion. Only a point should be better described. Why do not continue to affectionate f3 using other strategies?
We appreciate the reviewer´s comments. The results presented in this work are preliminary but solid, which lead us to try to answer any questions and, importantly, compel us to continue for the search of the molecules involved in the biological activities described here. We are doing some experiments to know more about the chemical composition of the fraction tested and the three of them. Of course, this is complicated and expensive. However, we are in the process of analyzing results and preparing another paper. We believe that this work, along with a previous one on other species, may encourage other researchers to start looking for calcium channel modulators in Cnidaria, which have not been reported or explored to date. Nevertheless, some changes were made to the manuscript to clarify and extend some aspects.

Reviewer 4 Report
The authors describe the effect the effect of a low molecular weight fraction from the venom of the sea anemone Lebrunia neglecta on boar sperm cells and HVA calcium channels from rat chromaffin cells. They provide credible proof that the components of this fraction impair boar sperm cells motility and and capacitation. Furthermore, they provide evidence that these components inhibit HVA calcium currents in chromaffin cells without altering the current kinetics and voltage relationship. Their study clearly shows the existence of Cav2+-channel toxins in cnidarian venom and extends the possible application of sea anemone toxins as pharmacological tools.
The research is carried out well and the presentation is generally clear, however, certain issues need to be addressed to improve the manuscript.
General comments
The text contains several typing errors, omitted or inserted words and style weaknesses or inconsistencies. Proofreading is highly recommended.
Specific comments are provided below and follow the sections of the manuscript.
Results and discussion
2.2. Sperm viability
The presentation of the results on sperm viability could be improved. A bar graph (treatments vs. control) might be a better way to present these results more clearly. Please provide statistics.
2.3. Sperm motility
Line 108: Table 2 title could be more precise, e.g. Sperm cells motility and agglutination after 4 hours of incubation with F3.
Line 113: replace “nule” for “zero”
Lines 119-122: The sentence “The origin of this agglutination could be explained by the presence of antigens in the sperm cell membrane produced by contact with the peptides contained in the venom fraction since the surface of spermatozoa is rich in glycoproteins [26].” is not very clear. Please elaborate.
Line 125: What is meant by chemical characterization?
Line 128: mg should be replaced with μg?
2.3. Capacitation
Line 129: Section numbering should be 2.4. All subsequent section numberings are therefore wrong as well and should be corrected.
Table 3: Please provide statistics.
2.6. F3 does not inhibit L-type calcium channels only
Lines 216-218: The authors claim that the L-type calcium channels are not the only target of the toxic components in the F3 venom fraction. They base their conclusion on the result where, when the L-type calcium channel blocker Nif and F3 were applied together, the current amplitude was reduced even more compared to Nif alone. This statement could only be true in the case where the application of Nif alone (at the concentration used in the experiments which is not specified anywhere) elicited a maximum inhibition of these channels. This, however, is not clearly demonstrated in the presented results. Proof should be provided that Nif alone completely blocked the L-type channel current. Otherwise, this section should be revised and the results interpreted accordingly.
Figure 5D: Label above the green line should be “Nif + F3”?
Materials and methods
In some sections, the description of methods is not sufficient to reproduce the results, nor are appropriate references included. Details below. Furthermore, the use of liter (l or L) is inconsistent throughout the manuscript text and figures. Please correct this.
3.1. Venom extraction and fractionation
This section lacks a more detailed description of extraction and chromatography procedures and conditions used. Please revise and include more details or provide a suitable reference.
3.4. Collection and semen sample preparation
This section lacks a more detailed description of the motility and viability assays. It also completely lacks the description of the sperm cells agglutination assay. Please revise this section and include the necessary details or properly provide suitable references if applicable.
3.5. Capacitation
Line 260: What is “750 μM CTC buffer”? The authors probably meant 750 μM CTC. Please check and provide correct details (and buffer used).
Lines 256 and 267: Are “capacitation medium” and “capacitation buffer” the same thing? If so, please use consistent naming, otherwise provide correct details for “capacitation buffer” in section 3.6. Toxin exposure (Line 267).
3.8. Electrophysiological recording
This section does not mention the use of Nif. Please provide details (concentration used) and provide the full name and abbreviation used in this manuscript (nifedipine – Nif) when the name is used for the first time.
Author Response
Reviewer #4
- The presentation of the results on sperm viability could be improved. A bar graph (treatments vs. control) might be a better way to present these results more clearly. Please provide statistics.
The table has been changed into a bar graph as suggested by the reviewer. We apologize for ommiting the statistics, this information has also been added.
- Line 108: Table 2 title could be more precise, e.g. Sperm cells motility and agglutination after 4 hours of incubation with F3.
The correction has been done as suggested.
- Line 113: replace “nule” for “zero”
This has been done as suggested.
- Lines 119-122: The sentence “The origin of this agglutination could be explained by the presence of antigens in the sperm cell membrane produced by contact with the peptides contained in the venom fraction since the surface of spermatozoa is rich in glycoproteins [26].” is not very clear. Please elaborate.
The sperm cells contain specific receptors (that contain carbohydrates) for agglutination factors in their membrane surfaces. This has been mainly studied for the sperm agglutination factors of some pathogenic bacteria in the reproductive tracts. However, since we still do not have enough chemical information about the components of our fraction, we cannot infer more about the mechanism of agglutination here.
- Line 125: What is meant by chemical characterization?
By chemical characterization we mean the elucitation of partial or complete chemical structure of all or some peptides in the fraction that could elicit the biological activities described in the study.
- Line 128: mg should be replaced with μg?
This has been corrected in the text.
- Line 129: Section numbering should be 2.4. All subsequent section numberings are therefore wrong as well and should be corrected.
This has been corrected.
- Table 3: Please provide statistics.
This information has been added.
- Lines 216-218: The authors claim that the L-type calcium channels are not the only target of the toxic components in the F3 venom fraction. They base their conclusion on the result where, when the L-type calcium channel blocker Nif and F3 were applied together, the current amplitude was reduced even more compared to Nif alone. This statement could only be true in the case where the application of Nif alone (at the concentration used in the experiments which is not specified anywhere) elicited a maximum inhibition of these channels. This, however, is not clearly demonstrated in the presented results. Proof should be provided that Nif alone completely blocked the L-type channel current. Otherwise, this section should be revised and the results interpreted accordingly.
Nifedipine is a selective blocker of L-type calcium channels. Chromaffin cells possess aproximately 50% of nifedipine-sensitive channels, which is consistent with the 54.7 ± 2.2 % inhibition we reported in line 251. The concentration used in the experiments was 5 µM, the maximum concentration for CaV channels (Fox, Nowycky and Tsien, 1987; Helton et al, 2005; Kitamura et al, 1997; Wang et al, 2018). Since the co-application of F3 with Nif results in a higher blockade than the one that can be elicited by Nif alone, we can conclude that F3 not only inhibits L-type current, but another target as well. Figure 6A shows the blockade values elicited by F3 alone and Nif alone; and figure 6C shows the response of the co-application of Nif+F3.
The correction about the concentration has been made in the text. The citations are enlisted at the end of the responses. We didn´t consider necessary to add them in the manuscript.
- Figure 5D: Label above the green line should be “Nif + F3”?
As it can be observed in Fig 6D, nifedipine was applied steadily prior, in the presence and over-lasting the application of F3. Therefore, yes, that is the right label. The green line indicates F3 superfusion over the nifedipine application.
- In some sections, the description of methods is not sufficient to reproduce the results, nor are appropriate references included. Details below. Furthermore, the use of liter (l or L) is inconsistent throughout the manuscript text and figures. Please correct this.
The use of “L” has been homogeneized.
- Venom extraction and fractionation. This section lacks a more detailed description of extraction and chromatography procedures and conditions used. Please revise and include more details or provide a suitable reference.
Some detailes about these procedures have been added as suggested.
- Collection and semen sample preparation.This section lacks a more detailed description of the motility and viability assays. It also completely lacks the description of the sperm cells agglutination assay. Please revise this section and include the necessary details or properly provide suitable references if applicable.
This information has been added in the methods section.
- Line 260: What is “750 μM CTC buffer”? The authors probably meant 750 μM CTC. Please check and provide correct details (and buffer used).
We apologize for the mistake, this has been corrected in the text.
- Lines 256 and 267: Are “capacitation medium” and “capacitation buffer” the same thing? If so, please use consistent naming, otherwise provide correct details for “capacitation buffer” in section 3.6. Toxin exposure (Line 267).
This has been corrected in the text.
- Electrophysiological recording. This section does not mention the use of Nif. Please provide details (concentration used) and provide the full name and abbreviation used in this manuscript (nifedipine – Nif) when the name is used for the first time.
The correction has been done. The full name and abbreviation was added in results section and the concentration was specified as suggested by the reviewer.
References
Fox AP, Nowycky MC, Tsien RW. Kinetic and pharmacological properties distinguishing three types of calcium currents in chick sensory neurones. J Physiol. 1987 Dec;394:149-72. doi: 10.1113/jphysiol.1987.sp016864. PMID: 2451016; PMCID: PMC1191955.
Helton TD, Xu W, Lipscombe D. Neuronal L-type calcium channels open quickly and are inhibited slowly. J Neurosci. 2005 Nov 2;25(44):10247-51. doi: 10.1523/JNEUROSCI.1089-05.2005. PMID: 16267232; PMCID: PMC6725800.
Kitamura, N., Ohta, T., Ito, S. et al. Calcium channel subtypes in porcine adrenal chromaffin cells. Pflügers Arch 434, 179–187 (1997). https://doi.org/10.1007/s004240050381
Wang Y, Tang S, Harvey KE, Salyer AE, Li TA, Rantz EK, Lill MA, Hockerman GH. Molecular Determinants of the Differential Modulation of Cav1.2 and Cav1.3 by Nifedipine and FPL 64176. Mol Pharmacol. 2018 Sep;94(3):973-983. doi: 10.1124/mol.118.112441. Epub 2018 Jul 6. PMID: 29980657.

Round 2
Reviewer 2 Report
Comments and suggestions have been given in the first round of review.
Author Response
- MALDI mass analysis of F3 indicates the presence of many components thus it is recommended to purify and characterize the component responsible for activity.
Yes, we agree with the reviewer that the purification and characterization of the compound or compounds responsible for the biological activity will be of great interest and must be done. We are in the process of obtaining data about the chemical structure of the components in the fractions. This information will be published in another paper.
- Is the activity specific to Cav1.2 or Cav1.3. This is very important for calcium channel pharmacology if the component/s responsible for activity is selective of specific to one Cav subtype since very few known compounds exhibit such specificity/selectivity.
As shown in Fig. 5, F3 affects a fraction of the nifedipine-sensitive CaV and an additional fraction of the total Cd2+ sensitive voltage-gated CaV current; therefore, only a non-specific action of F3 can be supported until this stage of the experiments. Future experiments should be performed using purified peptides from F3 to support a specific CaV target.
Reviewer 4 Report
The authors have adequately addressed the majority of the raised issues; however, certain concerns remain and are presented below.
1. Figure 3, line 165: mg has not been corrected to μg.
2. Authors’ answer: Nifedipine is a selective blocker of L-type calcium channels. Chromaffin cells possess aproximately 50% of nifedipine-sensitive channels, which is consistent with the 54.7 ± 2.2 % inhibition we reported in line 251. The concentration used in the experiments was 5 μM, the maximum concentration for CaV channels (Fox, Nowycky and Tsien, 1987; Helton et al, 2005; Kitamura et al, 1997; Wang et al, 2018). Since the co-application of F3 with Nif results in a higher blockade than the one that can be elicited by Nif alone, we can conclude that F3 not only inhibits L-type current, but another target as well. Figure 6A shows the blockade values elicited by F3 alone and Nif alone; and figure 6C shows the response of the co-application of Nif+F3.
Comment: Thank you for clearing up the question regarding Nif and its ability to maximally block this type of current at the concentration used. However, I suggest this fact be mentioned briefly in the results/discussion to make it clear for other readers as well.
Furthermore, the 54.7 ± 2.2 % inhibition refers to the Nif + F3 conditions (Line 253), while that of Nif alone is 40.5 ± 4.7 % (Line 248). Therefore, the above statement in the authors’s answer doesn’t seem entirely correct.
3. Line 299: 100X – is this magnification or objective used? Please specify.
4. The authors have corrected several typing errors and other mistakes in the text, however, the manuscript still contains many such errors and style weaknesses/inconsistencies.
To name but a few:
- Therapeutics should be therapeutic (Line 23)
- ml vs mL (Line 134)
- Lines 138-139: …have also been demonstrated… (missing been)
- the entire paragraph (Lines 143-150) is poorly written
- Line 182: …one of the purposes… (missing of)
- Line 189: Figure 4D should be properly cited in the text
- Line 243: …is appropriate to study..
- Inconsistent use of the multiplication sign (118 x 1.5 cm, 100 X, 400X, 80X106, etc.), spaces (1mg/mL, 2 mg/mL), g-force (180 g, 600g), etc.
- Lines 308-310: omitted spaces, wrong use of parentheses
The above list is just an example and is by no means complete. The entire text should be thoroughly checked again.
Author Response
- Figure 3, line 165: mg has not been corrected to μg.
Answer: Done.
- Authors’ answer: Nifedipine is a selective blocker of L-type calcium channels. Chromaffin cells possess aproximately 50% of nifedipine-sensitive channels, which is consistent with the 54.7 ± 2.2 % inhibition we reported in line 251. The concentration used in the experiments was 5 μM, the maximum concentration for CaV channels (Fox, Nowycky and Tsien, 1987; Helton et al, 2005; Kitamura et al, 1997; Wang et al, 2018). Since the co-application of F3 with Nif results in a higher blockade than the one that can be elicited by Nif alone, we can conclude that F3 not only inhibits L-type current, but another target as well. Figure 6A shows the blockade values elicited by F3 alone and Nif alone; and figure 6C shows the response of the co-application of Nif+F3.
Comment: Thank you for clearing up the question regarding Nif and its ability to maximally block this type of current at the concentration used. However, I suggest this fact be mentioned briefly in the results/discussion to make it clear for other readers as well.
Furthermore, the 54.7 ± 2.2 % inhibition refers to the Nif + F3 conditions (Line 253), while that of Nif alone is 40.5 ± 4.7 % (Line 248). Therefore, the above statement in the authors’s answer doesn’t seem entirely correct.
Answer: Thank you for the observation. We added in the results section a line to specify the nifidipine concentration employed in the experiments and specify that this is the maximal concentration for L-type channels inhibition. It was a mistake in the % inhibition, in the previous answer. It is true, 40.5% corresponds to nifedipine alone while 54.7% is for Nif + F3.
- Line 299: 100X – is this magnification or objective used? Please specify.
Answer: It is the magnification. Also added in text in the paper.
- The authors have corrected several typing errors and other mistakes in the text, however, the manuscript still contains many such errors and style weaknesses/inconsistencies.
To name but a few:
- Therapeutics should be therapeutic (Line 23)
- ml vs mL (Line 134)
- Lines 138-139: …have also been demonstrated… (missing been)
- the entire paragraph (Lines 143-150) is poorly written
- Line 182: …one of the purposes… (missing of)
- Line 189: Figure 4D should be properly cited in the text
- Line 243: …is appropriate to study..
- Inconsistent use of the multiplication sign (118 x 1.5 cm, 100 X, 400X, 80X106, etc.), spaces (1mg/mL, 2 mg/mL), g-force (180 g, 600g), etc.
- Lines 308-310: omitted spaces, wrong use of parentheses
The above list is just an example and is by no means complete. The entire text should be thoroughly checked again.
Answer: A full revision was done, thank you for your valuable comments to improve this manuscript.